# Robust Estimation of Neural Signals in Calcium Imaging

**Hakan Inan** [1]
inanh@stanford.edu

**Murat A. Erdogdu** [2,3]
erdogdu@cs.toronto.edu

**Mark J. Schnitzer** [1,4]
mschnitz@stanford.edu

[1]Stanford University  [2]Microsoft Research  [3]Vector Institute  [4]Howard Hughes Medical Institute

## Abstract

Calcium imaging is a prominent technology in neuroscience research which allows for simultaneous recording of large numbers of neurons in awake animals. Automated extraction of neurons and their temporal activity from imaging datasets is an important step in the path to producing neuroscience results. However, nearly all imaging datasets contain gross contaminating sources which could originate from the technology used, or the underlying biological tissue. Although past work has considered the effects of contamination under limited circumstances, there has not been a general framework treating contamination and its effects on the statistical estimation of calcium signals. In this work, we proceed in a new direction and propose to extract cells and their activity using robust statistical estimation. Using the theory of M-estimation, we derive a minimax optimal robust loss, and also find a simple and practical optimization routine for this loss with provably fast convergence. We use our proposed robust loss in a matrix factorization framework to extract the neurons and their temporal activity in calcium imaging datasets. We demonstrate the superiority of our robust estimation approach over existing methods on both simulated and real datasets.

## 1   Introduction

Calcium imaging has become an indispensable tool in systems neuroscience research. It allows simultaneous imaging of the activity of very large ensembles of neurons in awake and even freely behaving animals [3, 4, 6]. It relies on fluorescence imaging of intracellular calcium activity reported by genetically encoded calcium indicators. A crucial task for a neuroscientist working with calcium imaging is to extract signals (i.e. temporal traces and spatial footprints of regions of interest) from the imaging dataset. This allows abstraction of useful information from a large dataset in a highly compressive manner, losing little to no information. Automating this process is highly desirable, as manual extraction of cells and their activities in large-scale datasets is prohibitively laborious, and prone to flawed outcomes.

A variety of methods have been proposed for automated signal extraction in calcium imaging datasets, including the ones based on matrix factorization [13, 14, 15, 16], and image segmentation [1, 10]. Some of these tools were tailored to two-photon calcium imaging, for which signal-to-noise ratio is typically high, and the fluorescence background is fairly stable [3], whereas some targeted one-photon and microendoscopic calcium imaging [4, 5], which are often characterized by low SNR and large background fluctuations. Interestingly, least squares estimation has been a predominant paradigm among previous methods; yet there is no previous work addressing statistically the generic nature of calcium imaging datasets, which includes non-gaussian noise, non-cell background activity (e.g. neuropil), and overlapping cells not captured by algorithms (out-of-focus or foreground). As a consequence, the impact of such impurities inherent in calcium imaging on the accuracy of extracted signals has not been thoroughly investigated previously. This lack of focus on signal accuracy is worrisome as cell extraction is a fairly early step in the research pipeline, and flawed signals may lead to incorrect scientific outcomes.

In this work, we propose an approach which takes into account the practical nature of calcium imaging, and solves the signal extraction problem through robust estimation. First, we offer a mathematical abstraction of imaging datasets, and arrive at an estimator which is minimax robust, in the sense that is prevalent in the field of robust estimation. We then use this M-estimator to solve a matrix factorization problem, jointly yielding the temporal and spatial components of the extracted signals.

The main insight behind our robust estimation framework is that the signals present in imaging data are the superposition of many positive amplitude sources, and a lower amplitude noise component which could be well modeled by a normal distribution. That the majority of the components is positive stems from the fact that the underlying signals in calcium imaging are all made up of photons, and they elicit activity above a baseline as opposed to fluctuating around it. However, not all positive sources are cells that could be extracted by an algorithm (some could be neuropil, other noise, or non-captured cells); hence we model them as generic gross non-negative contamination sources. By using the machinery of robust estimation [7], we propose an M-estimator which is asymptotically minimax optimal for our setting.

We also propose a fast fixed-point optimization routine for solving our robust estimation problem. We show local linear convergence guarantees for our routine, and we demonstrate numerically that it converges very fast while only having the same per-step cost with gradient descent. The fast optimizer allows for very fast automated cell extraction in large-scale datasets. Further, since the final form for our loss function is simple and optimization only depends on matrix algebra, it is highly amenable to GPU implementation providing additional improvements.

We validate our robust estimation-based cell extraction algorithm on both synthetic and real datasets. We show that our method offers large accuracy improvements over non-robust techniques in realistic settings, which include classical scenarios such as overlapping cells and neuropil contamination. Particularly, our method significantly outperforms methods with non-robust reconstruction routines in metrics such as signal fidelity and crosstalk, which are crucial for steps subsequent to cell extraction.

## 2 M-Estimation under Gross Non-negative Contamination

In this section, we introduce our signal estimation machinery, based on the literature of robust M-estimation. The theory of M-estimation is well-developed for symmetric and certain asymmetric contamination regimes [2, 7, 9, 12]; however the existing theory does not readily suggest an optimal estimator suitable for finding the kind of signals present in fluorescence imaging of calcium in the brain. We first motivate and introduce a simple mathematical abstraction for this new regime, and then derive a minimax optimal M-estimator.

### 2.1 Noise Model & Mathematical Setting

For simplicity, we consider the setting of location estimation, which straightforwardly generalizes to multivariate regression.

Considering the nature of contamination in calcium imaging datasets, we base our noise model on the following observation: The signal background is dominated by the baseline activity which is well modeled by a normal distribution. This type of noise stems from the random arrivals of photons from the background in the imaging setup governed by a poisson process; this distribution very rapidly converges to a normal distribution. However, the signal background also contains other sources of noise such as neuropil activity, out-of-focus cells, and residual activity of overlapping cells not accounted for by the cell extraction method. The latter kind of contamination is very distinct from a normal-type noise; it is non-negative (or above the signal baseline), its characteristics are rather irregular and it may take on arbitrarily large values.

Consequently, we model the data generation through an additive noise source which is normally distributed $1 - \epsilon$ fraction of the time, and free to be any positive value greater than a threshold otherwise:

$$y_i = \beta^* + \sigma_i \tag{1}$$

$$\sigma_i \sim \begin{cases} \mathcal{N}(0,1), & \text{w.p. } 1 - \epsilon \\ H_\alpha, & \text{w.p. } \epsilon \end{cases} \tag{2}$$

$$H_\alpha \in \mathcal{H}_\alpha = \{\text{All distributions with support } [\alpha, \infty)\}, \quad \alpha \geq 0.$$

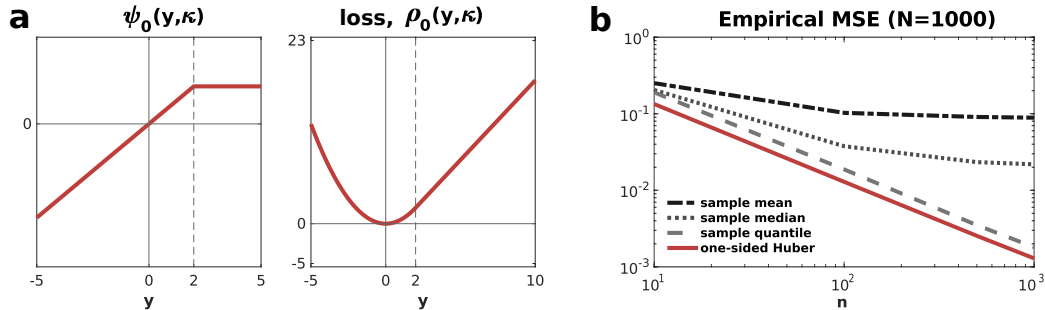

Figure 1: One-sided Huber. **(a)** loss function of one-sided Huber ($\rho$) and its derivative ($\psi$) for $\kappa = 2$. **(b)** One-sided Huber yields lower MSE compared to other known M-estimators under the distribution which causes the worst-case variance for any given estimator (for $\epsilon = 0.1$).

In above, $\beta^*$ is the true parameter, and is corrupted additively as in (1); $\sigma_i$ is a standard normal with $1 - \epsilon$ probability, and distributed according to an unknown distribution $H_\alpha$ with probability $\epsilon$. In the spirit of full generality, we allow $H_\alpha$ to be any probability distribution with support greater than a set value $\alpha$; particularly, it could be nonzero at arbitrarily large values. Therefore, $\epsilon$ could be interpreted as the gross contamination level. The parameter $\alpha$ could be interpreted as the minimum observed value of the positive contamination, although its exact value is insignificant outside our theoretical analysis. We denote the full noise distribution by $F_{H_\alpha}$, subscripted by $H_\alpha$.

Given the observations $\{y_i\}_{i=1}^n$, we estimate the true parameter $\beta^*$ with $\hat{\beta}$ by considering an equivariant M-estimator as follows

$$\hat{\beta} = \operatorname*{argmin}_{\beta} \sum_{i=1}^{n} \rho(y_i - \beta). \tag{3}$$

Typically, M-estimators are characterized by $\psi \triangleq \rho'$. In this paper, we are going to consider $\psi$'s with specific properties that allow for efficient optimization and more general theoretical guarantees. Let's define a set $\Psi = \{\psi \mid \psi \text{ is non-decreasing}\}$. If we choose an estimator $\psi \in \Psi$, finding a point estimate $\hat{\beta}$ through (3) becomes equivalent to solving the first order condition:

$$\sum_{i=1}^{n} \psi(y_i - \hat{\beta}) = 0. \tag{4}$$

This is simply because the members of $\Psi$ correspond to convex loss functions. Our focus is on such functions since they are typically easier to optimize, and offer global optimality guarantees.

## 2.2 One-Sided Huber Estimator and its Asymptotic Minimax Optimality

We are interested in finding an M-estimator for our noise model which is robust to the variation in the noise distribution ($H_\alpha$ in particular) in the sense of minimizing the worst-case deviation from the true parameter, as measured by the mean squared error. We first introduce our proposed estimator, and then show that it is exactly optimal in the aforementioned minimax sense.

**Definition 1** (*One-sided Huber*). *Define an estimator $\psi_0$ as follows:*

$$\psi_0(y, \kappa) = \begin{cases} y, & \text{if } y < \kappa \\ \kappa, & \text{if } y \geq \kappa, \end{cases} \tag{5}$$

*where $\kappa$ is defined in terms of the contamination level, $\epsilon$, according to*

$$\Phi(\kappa) + \frac{g(\kappa)}{\kappa} = \frac{1}{(1 - \epsilon)},$$

*with $\Phi(\cdot)$ and $g(\cdot)$ denoting the distribution and the density functions for a standard normal variable, respectively.*

We shall refer to $\psi_0$ as *one-sided Huber*, and denote with $\rho_0(\cdot, \kappa)$ its loss function (see Figure 1 for visualization). Clearly, $\psi_0 \in \Psi$, and therefore the loss function $\rho_0$ is convex. Under the data generation model introduced in the previous section, we can now state an asymptotic minimax result for $\psi_0$.

---
**Algorithm 1** Fast Solver for one-sided Huber Loss
---
**function** fp_solve($\mathbf{X}, \mathbf{Y}, k, \delta$)      //   $X = [x_1, \ldots, x_n]^T, Y = [y_1, \ldots, y_n]^T$

    1. **Compute:** $\mathbf{X}^+ = (\mathbf{X}^T \mathbf{X})^{-1} \mathbf{X}^T$, $\boldsymbol{\beta}^{LS} = \mathbf{X}^+ \mathbf{Y}$

    2. Initialize $\boldsymbol{\beta}^{(0)}$ at random, set $t = 0$.

    3. **while** $\left\| \boldsymbol{\beta}^{(t+1)} - \boldsymbol{\beta}^{(t)} \right\|_2 \geq \delta$ **do**

        $\boldsymbol{\beta}^{(t+1)} = \boldsymbol{\beta}^{LS} - \mathbf{X}^+ \max(0, \mathbf{Y} - \mathbf{X}\boldsymbol{\beta}^{(t)} - \kappa)$

        $t \leftarrow t + 1$.

    4. **end while**

**return** $\boldsymbol{\beta}^{(t)}$.

---

**Proposition 2.1.** *One-sided Huber $\psi_0$ yields an asymptotically unbiased M-estimator for $\mathcal{F}_{H_\kappa} = \{(1 - \epsilon)\Phi + \epsilon H_\kappa\}$. Further, $\psi_0$ minimizes the worst case asymptotic variance in $\mathcal{F}_{H_\kappa}$ , i.e.*

$$\psi_0 = \arg \inf_{\psi \in \Psi} \sup_{F \in \mathcal{F}_{H_\kappa}} V(\psi, F).$$

A proof for Proposition 2.1 is given in the supplementary material. Proposition 2.1 establishes that that one-sided Huber estimator has zero bias as long as the non-zero contamination is sufficiently larger than zero, and it also achieves the best worst-case asymptotic variance.

We would like to offer a discussion for a comparison between one-sided Huber and some other popular M-estimators, such as the sample mean ($\ell_2$ loss), the sample median ($\ell_1$ loss), Huber [7], and the sample quantile. First of all, the sample mean, the sample median, and Huber estimators all have symmetric loss functions and therefore suffer from bias. This is particularly detrimental for the sample mean and leads to unbounded MSE as the gross contamination tends to very large values. The bias problem may be eliminated using a quantile estimator whose quantile level is set according to $\epsilon$. However, this estimator has higher asymptotic variance than the one-sided Huber. We present in Figure 1b comparison of empirical mean square errors for different estimators under the noise distribution which causes the worst asymptotic variance among distributions in $\mathcal{F}_{\mathcal{H}_\kappa}$[1]. The MSEs of the sample mean and the sample median quickly become dominated by their bias with increasing $n$[2]. Although the quantile estimator was set up to be unbiased, its MSE (or equivalently, variance) is greater than the one-sided Huber. These results corroborate the theoretical properties of one-sided Huber, and affirm it as a good fit for our setting.

Although we have not come across a previous study of one-sided Huber estimator in this context, we should note that it is related to the technique in [11], where samples are assumed to be nonnegative, and in the sample mean estimator summands are shrunk when they are above a certain threshold (this technique is called winsorizing). However, their model and application are quite different than what we consider in this paper.

### 2.3   Generalization to Regression Setting

Here we introduce the regression setting which we will use for the remainder of the paper. We observe $\{y_i, \mathbf{x}_i\}_{i=1}^n$, where $\mathbf{x}_i \in \mathbb{R}^p$ could be either fixed or random, and $y_i$'s are generated according to $y_i = \langle \mathbf{x}_i, \boldsymbol{\beta}^* \rangle + \sigma_i^g + \sigma_i^h$, where $\boldsymbol{\beta}^* \in \mathbb{R}^p$ is the true parameter, and $\sigma_i^h$ and $\sigma_i^g$ are as previously defined. We estimate $\boldsymbol{\beta}^*$ with

$$\hat{\boldsymbol{\beta}} = \underset{\boldsymbol{\beta}}{\operatorname{argmin}} f_\kappa(\beta) := \sum_{i=1}^n \rho_0(y_i - \langle \mathbf{x}_i, \boldsymbol{\beta} \rangle, \kappa). \tag{6}$$

Classical M-estimation theory establishes –under certain regularity conditions– that the minimax optimality in Section 2.2 carries over to regression; we refer reader to [8] for details.

## 3   Fast Fixed-point Solver for One-Sided Huber Loss

We are interested in solving the robust regression problem in (6) in the large-scale setting due to the large field of view and length of most calcium imaging recordings. Hence, the solver for our problem

**Algorithm 2** Tractable and Robust Automated Cell Extraction

---

**function** EXTRACT($\mathbf{M}, N, \kappa, \delta$)

    1. Initialize $\mathbf{S}^{(0)}, \mathbf{T}^{(0)}$, set $t = 0$.

    2. **for** t=1 to N **do**

        $\mathbf{T}^{(t+1)} = \text{fp\_solve\_nonneg}(\mathbf{S}^{(t)}, \mathbf{M}, \kappa, \delta)$

        $\mathbf{S}^{(t+1)} = \text{fp\_solve\_nonneg}(\mathbf{T}^{(t)^T}, \mathbf{M}^T, \kappa, \delta)^T$

        $\mathbf{S}^{(t+1)}, \ \mathbf{T}^{(t+1)} = \text{remove\_redundant}\left(\mathbf{S}^{(t+1)}, \mathbf{T}^{(t+1)}\right)$

    3. **end for**

**return** $\mathbf{S}^{(t)}, \ \mathbf{T}^{(t)}$.

---

should ideally be tractable for large $n$ and also give as accurate an output as possible. To this end, we propose a fixed point optimization method (Algorithm 1), which has a step cost equal to that of gradient descent, while converging to the optimum at rates more similar to Newton's method. The following proposition establishes the convergence of our solver.

**Proposition 3.1.** *Let $\beta^*$ be the fixed point of Algorithm 1 for the problem (6), and let $\lambda_{max}$ and $\lambda_{min} > 0$ denote the extreme eigenvalues of $\sum_{i=1}^{n} x_i x_i^T$, and let $\max_i \|x_i\| \leq k$. Assume that for a subset of indeces $s \subset \{1, 2, ..., n\}$, $\exists \Delta_s > 0$ such that $y_i - \langle x_i, \beta^* \rangle \leq \kappa - \Delta_s$ and denote the extreme eigenvalues of $\sum_{i \in s} x_i x_i^T$ by $\gamma_{max}$ and $\gamma_{min} > 0$ satisfying $\lambda_{max}\gamma_{max}/\lambda_{min}^2 < 2$. If the initial point $\beta_0$ is close to the true minimizer, i.e., $\|\beta_0 - \beta^*\|_2 \leq k/\Delta_s$, then Algorithm 1 converges linearly,*

$$f_\kappa(\beta^t) - f_\kappa(\beta^*) \leq \left(1 - 2\frac{\gamma_{min}}{\lambda_{max}} + \frac{\gamma_{max}\gamma_{min}}{\lambda_{min}^2}\right)^t \left[f_\kappa(\beta^0) - f_\kappa(\beta^*)\right]. \tag{7}$$

A proof for Proposition 3.1 is given in the supplementary material.

Our solver is second order in nature[3], hence its convergence behavior should be close to that of Newton's method. However, there is one caveat: the second derivative of the one-sided Huber loss is not continuous. Therefore, one cannot expect to achieve a quadratic rate of convergence; this issue is commonly encountered in M-estimation. Nevertheless, Algorithm 1 converges very fast in practice.

We compare our solver to Newton's method and gradient descent by simulating a regression setting where we synthesize a 100 x 100 movie frame ($\mathbf{Y}$) with 100 neurons (see Section 5 for details). Then, given the ground truth cell images ($\mathbf{X}$), we optimize for the fluorescence traces for the single frame ($\boldsymbol{\beta}$) using the three algorithms. For our fixed-point solver, we use $\kappa = 1$. For gradient descent, we set the step size to the reciprocal of the largest eigenvalue of the hessian (while not taking into account the time taken to compute it). Results are shown in Figure 2. Our solver has close convergence behavior to that of Newton's method, while taking much less time to achieve the same accuracy due to its small per-step cost. We would like to also note that estimating the entire matrix of fluorescence traces (or cell images) does not require any modification of Algorithm 1; hence, in practice estimating entire matrices of components at once does not cause much computational burden. For Newton's method, every frame (or every pixel) requires a separate hessian; runtime in this case scales at least linearly.

## 4   Robust Automated Cell Extraction

We now introduce our proposed method for automated cell extraction via robust estimation. Our method is based on a matrix factorization framework, where we model the imaging data as the matrix product of a spatial and a temporal matrix with additive noise:

$$\mathbf{M} = \mathbf{ST} + \boldsymbol{\Sigma}.$$

In above, $\mathbf{M} \in \mathbb{R}^{d_S \times d_T}$ is the movie matrix, $\mathbf{S} \in \mathbb{R}_+^{d_S \times m}$ and $\mathbf{T} \in \mathbb{R}_+^{m \times d_T}$ are the nonnegative spatial and temporal matrices, respectively. $\boldsymbol{\Sigma} \in \mathbb{R}^{d_S \times d_T}$ is meant to model the normal noise corrupted with non-negative contamination, and $\boldsymbol{\Sigma}_{ij}$ has the same distribution with $\sigma$ in (2) (up to the noise standard deviation). Our main contribution in this work is that we offer a method which estimates $S$ and $T$ using the one-sided Huber estimator, which provides the optimal robustness against the non-negative contamination inherent in calcium imaging, as discussed in Section 2.

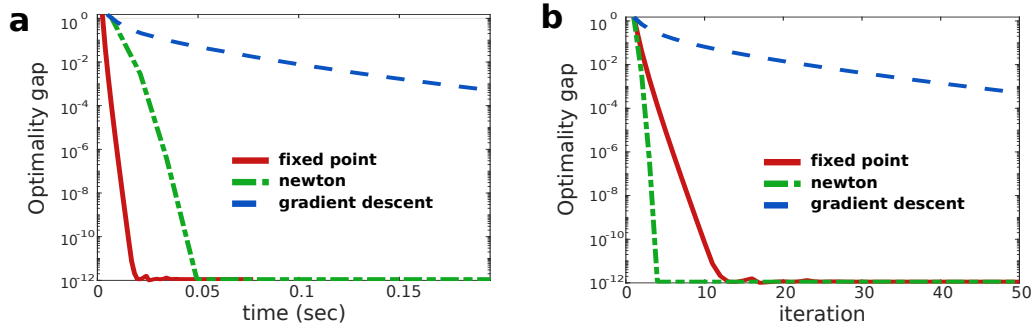

Figure 2: Our fixed point solver converges to the optimum with similar rates with Newton's method, while being more computationally efficient. **(a)** Optimality gap versus absolute time. **(b)** Optimality gap versus number of iterations. Fixed point solver achieves the same accuracy with a notably faster speed compared to Newton's method and gradient descent.

Our cell extraction algorithm starts by computing initial estimates for the matrices $\mathbf{S}$ and $\mathbf{T}$. This is done by (1) detecting a cell peak from the time maximum of the movie one cell at a time (2) solving for the current cell's spatial and the temporal components using the one-sided Huber estimator (3) repeating until a stopping criterion is reached. We detail this step in the supplementary material.

After initial guesses for $\mathbf{S}$ and $\mathbf{T}$ are computed, the main update algorithm proceeds in a straightforward manner, where multiple alternating robust regression steps are performed using the one-sided Huber loss. At each step, new estimates of $\mathbf{S}$ and $\mathbf{T}$ are computed based on $\mathbf{M}$ and the current estimate of the other matrix. For computing the estimates, we use the fast fixed-point algorithm derived in Section 3. However, since we constrain $\mathbf{S}$ and $\mathbf{T}$ to be nonnegative matrices, the fixed-point solver cannot be used without constraints that enforce non-negativity. To this end, we combine our solver with the alternating directions method of multipliers(ADMM), a dual ascent method which solves for multiple objectives by consensus. We call the combined solver `fp_solve_nonneg()`. Note that, due to the symmetry between the two alternating steps, we use the same solver for computing both $\mathbf{S}$ and $\mathbf{T}$.

We do minimal post-processing at the end of each step to remove redundant components. Specifically, we identify and remove near duplicate components in $\mathbf{S}$ or $\mathbf{T}$, and we then eliminate components which have converged to zero. We repeat these steps alternatingly for a desired number of steps $N$.

Selection of $\kappa$ depends on the positive contamination level; nevertheless, we have observed that precise tuning of $\kappa$ is not necessary in practice. A range of $[0.5, 1]$ times the standard deviation of the normally distributed noise is reasonable for $\kappa$ for most practices. One should note, however, that although the robust estimator has favorable mis-specification bias, it might become significant under crucially low SNR conditions. For instance, setting a small $\kappa$ in such cases will likely lead to detrimental under-estimation. On the other hand, setting high $\kappa$ values decreases the estimator robustness ( this makes the loss function approach the $\ell_2$ loss). Consequently, the advantage of robust estimation is expected to diminish in extremely low SNR regimes.

Our algorithm has a highly favorable runtime in practice owing to the simplicity of its form. Furthermore, since the solver we use relies on basic matrix operations, we were able to produce a GPU implementation, allowing for further reduction in runtime. Comparison of our GPU implementation to other algorithms in their canonical forms naturally causes bias; therefore, we defer our runtime comparison results to the supplementary material.

From here on, we shall call our algorithm EXTRACT.

## 5 Experiments

In this section, we perform experiments on both simulated and real data in order to establish the improved signal accuracy obtained using EXTRACT. We represent the signal accuracy with two quantities: (1) signal fidelity, which measures how closely a temporal (fluorescence trace) or spatial (cell image) signal matches its underlying ground truth, and (2) signal crosstalk, which quantifies interference from other sources, or noise. We primarily focus on temporal signals since they typically

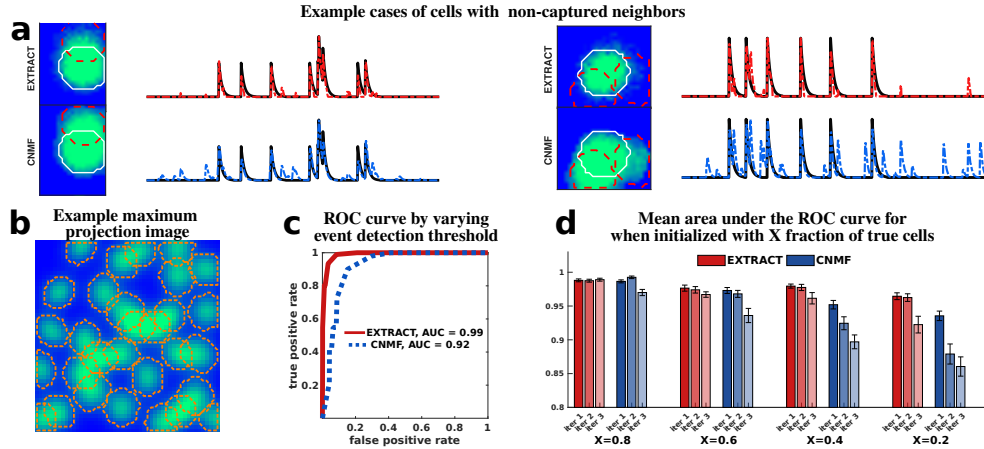

Figure 3: Performance comparison of EXTRACT vs. CNMF for movies with overlapping image sources. **(a)** Examples where a captured cell (circled in white) is overlapping with non-captured neighbors (circled in red). Ground truth traces are shown in black. EXTRACT finds images and traces that match closely with the ground truth, where CNMF admits notable crosstalk from neighbors both in its found cell images and traces.**(b)** An example maximum projection of an imaging movie in time. **(c)** An example ROC curve for X=0.4, computed by varying event detection threshold and averaging TPR and FPR over single cells for each threshold. **(d)** Mean area under the ROC curve computed over 20 experiments for each initial fraction of true cells, X, and each iteration. EXTRACT consistently outperforms CNMF, with the performance lead becoming significant for lower X. Error bars are 1 s.e.m.

represent the entirety of the calcium movie for the steps subsequent to cell extraction. As opposed to using simple correlation based metrics, we compute true and false positive detection rates based on estimated calcium events found via simple amplitude thresholding. We then present receiver operating characteristics (ROC) based metrics. We compare EXTRACT to the two dominantly used cell extraction methods: CNMF [15], and spatio-temporal ICA [13], the latter of which we will simply refer to as ICA. Both methods are matrix factorization methods like EXTRACT; CNMF estimates its temporal and spatial matrices alternatingly, and jointly estimates traces and its underlying calcium event peaks, and ICA finds a single unmixing matrix which is then applied to the singular value decomposition (SVD) of the movie to jointly obtain traces and images. CNMF uses quadratic reconstruction loss with $\ell_1$ penalty, whereas ICA uses a linear combination of movie data guided by high order pixel statistics for reconstruction; hence they both can be considered as non-robust estimation techniques.

**Simulated data.** For simulated movies, we use a field of view of size 50 by 50 pixels, and produce data with 1000 time frames. We simulate 30 neurons with gaussian shaped images with standard deviations drawn from $[3, 4.8]$ uniformly. We simulate the fluorescence traces using a Poisson process with rate $0.01$ convolved with an exponential kernel with a time constant of 10 frames. We corrupt the movie with independent and normally distributed noise whose power is matched to the power of the neural activity so that average pixel-wise SNR in cell regions is 1. We have re-run our experiments with different SNR levels in order to establish the independence of our key results from noise level; we report them in the supplementary material.

## 5.1 Crosstalk between cells for robust vs. non-robust methods

As a first experiment, we demonstrate consequences of a common phenomenon, namely cells with overlapping spatial weights. Overlapping cells do not pose a significant problem when their spatial components are correctly estimated; however, in reality, estimated images typically do not perfectly match their underlying excitation, or some overlapping cells might not even be captured by the extraction algorithm. In the latter two cases, crosstalk becomes a major issue, causing captured cells to carry false calcium activity in their fluorescence traces.

We try to reproduce the aforementioned scenarios by simulating movies, and initializing the algorithms of interest with a fraction of the ground truth cells. Our aim is to set up a controlled environment to (1) quantitatively investigate the crosstalk in the captured cell traces due to missing cells, (2) observe the effect of alternating estimation on the final accuracy of estimates. In this case, the outputs of

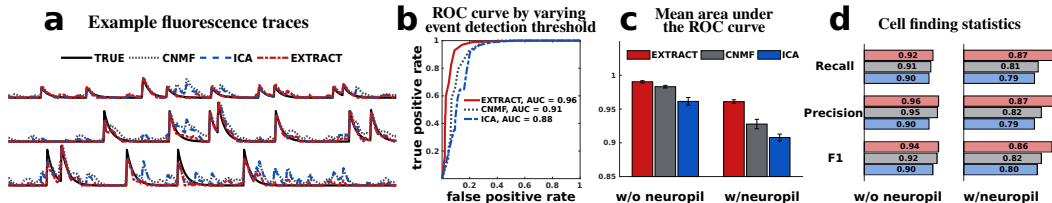

Figure 4: EXTRACT outperforms other algorithms in the existence of neuropil contamination. **(a)** Example traces from algorithm outputs overlaid on the ground truth traces. EXTRACT produces traces closest to the ground truth, admitting significantly less crosstalk compared to others. **(b)** An example ROC curve for an instance with neuropil. **(c)** Mean area under the curve computed over 15 experiments, and separately for with and without neuropil. EXTRACT shows better performance, and its performance is the most robust against neuropil contamination. **(d)** Average cell finding statistics over 15 experiments, computed separately for with and without neuropil. EXTRACT achieves better competitive performance especially when there is neuropil contamination.

alternating estimation algorithms should deteriorate through the iteration loop since they estimate their components based on imperfect estimates of each other. We select EXTRACT and CNMF for this experiment since they are both alternating estimation algorithms.

We initialize the algorithms with 4 different fractions of ground truth cells: $X = \{0.2, 0.4, 0.6, 0.8\}$. We carry out 20 experiments for each $X$, and we perform a 3 alternating estimation iterations for each algorithm. This number was chosen with the consideration that CNMF canonically performs 2 iterations on its initialized components. We report results for 6 iterations in the supplementary material. At the end of each iteration, we detect calcium events from the algorithms' fluorescence traces, and match them with the ground truth spikes to compute event true positive rate (TPR) and event false positive rate (FPR).

Figure 3 summarizes the results of this experiment. At the end of the 3 iterations, EXTRACT produces images and traces that are visually closer to ground truth in the existence of non-captured neighboring cells with overlapping images (Figure 3a). Figure 3c shows the ROC curve from one instance of the experiment, computed by varying the threshold amplitude for detecting calcium events, and plotting FPR against TPR for each threshold. We report quantitative performance by the area under the ROC curve (AUC). We average the AUCs over all the experiments performed for each condition, and report it separately for each iteration in Figure 3d. EXTRACT outperforms CNMF uniformly, and the performance gap becomes pronounced with very low fraction of initially provided cells. This boost in the signal accuracy over non-robust estimators (e.g. ones with quadratic penalty) stands to validate our proposed robust estimator and its underlying model assumptions.

## 5.2 Cell extraction with neuropil contamination

In most calcium imaging datasets, data is contaminated with non-cellular calcium activity caused by neuropil. This may interfere with cell extraction by contaminating the cell traces, and by making it difficult to accurately locate spatial components of cells. We study the effect of such contamination by simulating neural data and combining it with neuropil activity extracted from real two-photon imaging datasets. For this experiment, we use EXTRACT, CNMF and ICA.

In order for a fair comparison, we initialize all algorithms with the same set of initial estimates. We choose to use the greedy initializer of CNMF to eliminate any competitive advantage EXTRACT might have due to using its native initializer. We perform 15 experiments with no neuropil, and 15 with added neuropil. We match the variance of the neuropil activity to that of the gaussian noise while keeping SNR constant. For each experiment, we compute (1) cell trace statistics based on the ROC curve as previously described, (2) cell finding statistics based on precision, recall, and F1 metrics.

EXTRACT produces qualitatively more accurate fluorescence traces (Figure 4a), and it outperforms both CNMF and ICA quantitatively (Figure 4b,c), with the performance gap becoming more significant in the existence of neuropil contamination. Further, EXTRACT yields more true cells than the other methods with less false positives when there is neuropil (Figure 4d).

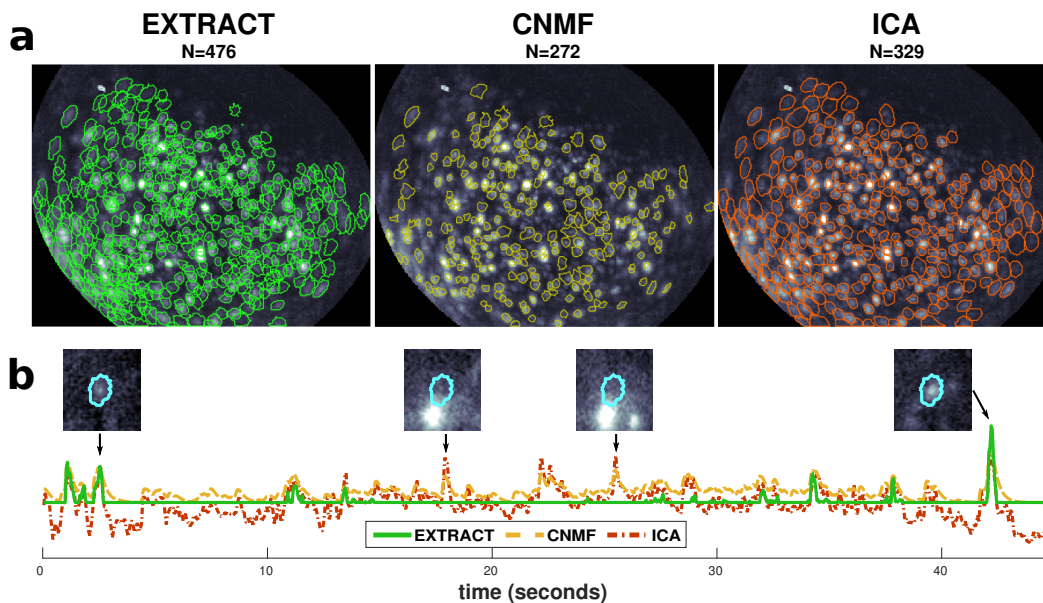

Figure 5: EXTRACT better estimates neural signals in microendoscopic single-photon imaging data. **(a)** The manually classified "good" cells for all 3 algorithms overlaid on the maximum of the imaging movie in time. Letter N refers to the total good cell count. **(b)** The fluorescence traces of the 3 algorithms belonging to the same cell. The cell has significantly low SNR compared to a neighbor cell which is also captured by all the methods. The time frames with arrows pointing to them are shown with the snapshot of the cell (circled in green) and its surrounding area. EXTRACT correctly assigns temporal activity to the cell of interest, while other algorithms register false calcium activity from the neighboring cell.

## 5.3 Cell extraction from microendoscopic single-photon imaging data

Data generated using microendoscopic single-photon calcium imaging could be quite challenging due to low SNR, and fluctuating background (out of focus fluorescence activity etc.). We put EXTRACT to test in this data regime, using an imaging dataset recorded from the dorsal CA1 region of the mouse hippocampus [17], an area known to have high cell density. We compare EXTRACT with CNMF and ICA. For this experiment, the output of each algorithm was checked by human annotators and cells were manually classified to be true cells or false positives judging from the match of their temporal signal to the activity in the movie.

EXTRACT successfully extracts the majority of the cells apparent in the maximum image of the movie in time dimension, and is able to capture highly overlapping cells (Figure 5a). EXTRACT also accurately estimates the temporal activity. Figure 5b shows an instance of a dim cell with a high SNR neighboring cell, both of which are captured by all three algorithms. While CNMF and ICA both falsely show activity when the neighbor is active, EXTRACT trace seems immune to this type of contamination and is silent at such instants.

## 6  Conclusion

We presented an automated cell extraction algorithm for calcium imaging which uses a novel robust estimator. We arrived at our estimator by defining a generic data model and optimizing its worst-case performance. We proposed a fast solver for our estimation problem, which allows for tractable cell extraction in practice. As we have demonstrated in our experiments, our cell extraction algorithm, EXTRACT, is a powerful competitor for the existing methods, performing well under different imaging modalities due to its generic nature.

## Acknowledgements

We gratefully acknowledge support from DARPA and technical assistance from Biafra Ahanonu, Lacey Kitch, Yaniv Ziv, Elizabeth Otto and Margaret Carr.

## Footnotes

[1] Refer to the proof of Proposition 2.1 for the form of this distribution.

[2] We omit Huber in this comparison since its MSE is also bias-dominated.

[3]Interested reader is referred to the supplementary material for a more rigorous argument.

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
