[Supplementary Material]

# Supplementary Material

## A Proof of Proposition 2.1

We're trying to prove the following proposition:

**Proposition A.1.** *One-sided Huber $\psi_0$ yields an unbiased M-estimator for $\mathcal{F} = (1 - \epsilon)\Phi + \epsilon H$ if and only if the contamination $H$ belongs to $\mathcal{H}_\kappa = \{H : H(x) = 0 \text{ for } x < \kappa \text{ and } I(H) < \infty\}$. Further, $\psi_0$ minimizes the worst case asymptotic variance in $\mathcal{F}$, i.e.*

$$\psi_0 = \arg \inf_{\psi \in \Psi} \sup_{F \in \mathcal{F}} V(\psi, F).$$

First, note that $F = (1 - \epsilon)\Phi + \epsilon H$ yields an unbiased M-estimator for $\psi_0$ if and only if

$$
\begin{aligned}
0 =& \mathbb{E}_F[\psi_0(y)] = (1 - \epsilon)\mathbb{E}_\Phi[\psi(y)] + \epsilon\mathbb{E}_H[\psi_0(y)], \\
=& (1 - \epsilon)\left\{\int_{-\infty}^\kappa y\phi(y)dy + \kappa\int_\kappa^\infty \phi(y)dy\right\} + \epsilon\left\{\int_{-\infty}^\kappa yh(y)dy + \kappa\int_\kappa^\infty h(y)dy\right\}, \\
=& (1 - \epsilon)\left\{-\phi(\kappa) + \kappa - \kappa\Phi(\kappa)\right\} + \epsilon\left\{\int_{-\infty}^\kappa yh(y)dy + \kappa\int_\kappa^\infty h(y)dy\right\}.
\end{aligned}
$$

Using $\Phi(\kappa) + \phi(\kappa)/\kappa = 1/(1 - \epsilon)$ for the first term on the right hand side, we obtain

$$0 = \int_{-\infty}^\kappa (\kappa - y)h(y)dy,$$

which is satisfied if and only if the support of $H$ is $[\kappa, \infty)$.

For the variance calculations, we use that one sided Huber estimator of $\psi_0$ is unbiased for the class of distributions $\mathcal{F} = (1 - \epsilon)\Phi + \epsilon\mathcal{H}_\kappa$. We calculate the variance for $\psi_0$ for some $F \in \mathcal{F}$ using $\mathbb{E}_F[\psi_0^2]/\mathbb{E}[\psi_0']^2$. The numerator can be written as

$$
\begin{aligned}
\mathbb{E}_F[\psi_0^2] =& (1 - \epsilon)\mathbb{E}_\Phi[\psi_0^2] + \epsilon\mathbb{E}_H[\psi_0^2], \\
=& (1 - \epsilon)\left\{\int_{-\infty}^\kappa y^2\phi(y)dy + \kappa^2\int_\kappa^\infty \phi(y)dy\right\} + \epsilon\kappa^2, \\
=& (1 - \epsilon)\left\{\Phi(\kappa) - \kappa\phi(\kappa) - \kappa^2\Phi(\kappa)\right\} + \kappa^2, \\
=& (1 - \epsilon)\Phi(\kappa).
\end{aligned}
$$

Similarly for the denominator, we write

$$
\mathbb{E}_F[\psi_0']^2 = \left\{(1 - \epsilon)\mathbb{E}_\Phi[\psi_0'] + \epsilon\mathbb{E}_H[\psi_0']\right\}^2,
$$
$$
= \left\{(1 - \epsilon)\int_{-\infty}^\kappa \phi(y)dy\right\}^2 = (1 - \epsilon)^2\Phi(\kappa)^2.
$$

Therefore the asymptotic variance is given as

$$V(\psi_0, F) = [(1 - \epsilon)\Phi(\kappa)]^{-1},$$

which is constant over the contamination class $\mathcal{H}_\kappa$.

Now, define a distribution $F_0$ by its density $f_0$ satisfying $-d\log(f_0)/dt = \psi_0$:

$$
f_0(y) = \begin{cases} (1 - \epsilon)\phi(y), & \text{if } y < \kappa \\ (1 - \epsilon)\phi(\kappa)\exp(-\kappa y + \kappa^2), & \text{if } y \geq \kappa \end{cases} \tag{1}
$$

First, we need to check whether $F_0 \in \mathcal{F}$. It is easy to check that $f_0$ (and the corresponding contamination) is a distribution, i.e. it integrates to 1 by the condition $\Phi(\kappa) + \phi(\kappa)/\kappa = 1/(1 - \epsilon)$. Then, $\forall F \in \mathcal{F}$, we have

$$V(\psi_0, F) = V(\psi_0, F_0). \tag{2}$$

Moreover, a straightforward application of Cauchy-Schwartz inequality yields

$$V(\psi, F_0) = \frac{\mathbb{E}_{F_0}[\psi^2]}{\mathbb{E}_{F_0}[\psi']^2} \geq \frac{1}{I(F_0)}$$

with equality only if $\psi \propto f_0'/f_0$. Combining this with the previous result, we obtain

$$\sup_{F \in \mathcal{F}} V(\psi_0, F) = V(\psi_0, F_0) = \inf_{\psi} V(\psi, F_0).$$

Finally, note that the left equality is in fact weaker than the statement in (2), and one can verify that the Fisher information is given as $I(F_0) = (1 - \epsilon)\Phi(\kappa)$.

## B  Proof of Proposition 3.1

We consider the following objective

$$\beta_* = \mathrm{argmin} f(\boldsymbol{\beta}) := \sum_{i=1}^{n} \rho_0(y_i - \langle \mathbf{x}_i, \beta \rangle, \kappa). \tag{3}$$

Setting:

- Assume that for some $S \subset [n]$ and $\Delta_S > 0$ such that $y_i - \langle \mathbf{x}_i, \boldsymbol{\beta}_* \rangle \leq \kappa - \Delta_S$ for $i \in S \subset [n]$. (Including more indeces in $S$ results in smaller $\Delta_S$).
- Let $\max_i \|\mathbf{x}_i\|_2 \leq k$.
- $\sum_{i \in S} \mathbf{x}_i \mathbf{x}_i^T \succeq \lambda_S \mathbf{I}$. This assumption is reasonable when $n$ is large and consequently there are many samples in the quadratic regime.
- $\lambda_{\max}$ and $\lambda_{\min}$ are the largest and smallest eigenvalues of $X^T X$, respectively.

For $\boldsymbol{\beta}$ in the ball centered around $\boldsymbol{\beta}_*$ with radius $\Delta_S/k$, we have for $\forall i \in S$,

$$\begin{aligned} y_i - \langle \mathbf{x}_i, \boldsymbol{\beta} \rangle &= y_i - \langle \mathbf{x}_i, \boldsymbol{\beta}_* \rangle + \langle \mathbf{x}_i, \boldsymbol{\beta}_* - \boldsymbol{\beta} \rangle \\ &\leq \kappa - \Delta_S + \|\mathbf{x}_i\|_2 \|\boldsymbol{\beta}_* - \boldsymbol{\beta}\|_2 \\ &\leq \kappa. \end{aligned}$$

Therefore, when the iterates $\boldsymbol{\beta}$ get close to the true minimizer, $\forall i \in S$, residual corresponding to sample $i$ falls into the quadratic region. This implies that the Hessian satisfies

$$\boldsymbol{\nabla}^2 f(\boldsymbol{\beta}) = \sum_{i \,:\, y_i - \langle x_i, \boldsymbol{\beta} \rangle \leq \kappa} \mathbf{x}_i \mathbf{x}_i^T \succeq \sum_{i \in S} \mathbf{x}_i \mathbf{x}_i^T \succeq \lambda_S \mathbf{I}, \tag{4}$$

which says that in the ball $B = \{\boldsymbol{\beta} \,:\, \|\boldsymbol{\beta} - \boldsymbol{\beta}_*\|_2 \leq \Delta/k\}$, the objective function $f$ is $\lambda_S$-strongly convex. Strong convexity implies smoothness, i.e., $\boldsymbol{\nabla}^2 f \preceq \gamma_S \mathbf{I}$ for $\forall \boldsymbol{\beta} \in B$. In this regime, the following calculation is standard.

Assuming that the current iterate is $\boldsymbol{\beta}$, our approach takes a step of the following form,

$$\boldsymbol{\beta}^+ = \boldsymbol{\beta} - (X^T X)^{-1} \nabla f(\boldsymbol{\beta}). \tag{5}$$

By $\gamma_S$-smoothness, we can write

$$\begin{aligned} f(\boldsymbol{\beta}^+) &\leq f(\boldsymbol{\beta}) + \langle \nabla f(\boldsymbol{\beta}), \boldsymbol{\beta}^+ - \boldsymbol{\beta} \rangle + \frac{\gamma_S}{2} \|\boldsymbol{\beta}^+ - \boldsymbol{\beta}\|_2^2 \\ &\leq f(\boldsymbol{\beta}) - \langle \nabla f(\boldsymbol{\beta}), (X^T X)^{-1} \nabla f(\boldsymbol{\beta}) \rangle + \frac{\gamma_S}{2} \|(X^T X)^{-1} \nabla f(\boldsymbol{\beta})\|_2^2 \\ &\leq f(\boldsymbol{\beta}) - \frac{1}{\lambda_{\max}} \|\nabla f(\boldsymbol{\beta})\|_2^2 + \frac{\gamma_S}{2\lambda_{\min}^2} \|\nabla f(\boldsymbol{\beta})\|_2^2 \\ &= f(\boldsymbol{\beta}) - \left\{ \frac{1}{\lambda_{\max}} - \frac{\gamma_S}{2\lambda_{\min}^2} \right\} \|\nabla f(\boldsymbol{\beta})\|_2^2. \end{aligned}$$

By $\lambda_S$-strong convexity

$$\begin{aligned} f(\boldsymbol{\beta}') &\geq f(\boldsymbol{\beta}) + \langle \nabla f(\boldsymbol{\beta}), \boldsymbol{\beta}' - \boldsymbol{\beta} \rangle + \frac{\lambda_S}{2} \|\boldsymbol{\beta}' - \boldsymbol{\beta}\|_2^2 \\ &\geq f(\boldsymbol{\beta}) - \frac{1}{2\lambda_S} \|\nabla f(\boldsymbol{\beta})\|_2^2. \end{aligned}$$

The second inequality follows from setting $\boldsymbol{\beta}' = \boldsymbol{\beta} - 1/\lambda_{\min}\nabla f(\boldsymbol{\beta})$ which is the minimizer of the right hand side of the first line. Choosing $\boldsymbol{\beta}' = \boldsymbol{\beta}_*$ above yields

$$\frac{1}{2\lambda_S}\|\nabla f(\boldsymbol{\beta})\|_2^2 \geq f(\boldsymbol{\beta}) - f(\boldsymbol{\beta}_*).$$

Using this and the smoothness inequality, we write

$$
\begin{aligned}
f(\boldsymbol{\beta}^+) - f(\boldsymbol{\beta}_*) \leq & f(\boldsymbol{\beta}) - f(\boldsymbol{\beta}_*) - \left\{\frac{1}{\lambda_{\max}} - \frac{\gamma_S}{2\lambda_{\min}^2}\right\}\|\nabla f(\boldsymbol{\beta})\|_2^2 \\
\leq & f(\boldsymbol{\beta}) - f(\boldsymbol{\beta}_*) - 2\lambda_S\left\{\frac{1}{\lambda_{\max}} - \frac{\gamma_S}{2\lambda_{\min}^2}\right\}(f(\boldsymbol{\beta}) - f(\boldsymbol{\beta}_*)) \\
= & \left\{1 - 2\frac{\lambda_S}{\lambda_{\max}} + \frac{\gamma_S\lambda_S}{\lambda_{\min}^2}\right\}(f(\boldsymbol{\beta}) - f(\boldsymbol{\beta}_*))
\end{aligned}
$$

This is linear convergence with coefficient $1 - 2\frac{\lambda_S}{\lambda_{\max}} + \frac{\gamma_S\lambda_S}{\lambda_{\min}^2}$ and the following condition must hold

$$\frac{\lambda_{\max}}{\lambda_{\min}^2} < \frac{2}{\gamma_S}.$$

# C   Properties and Empirical Performance of the Fixed-point Solver in Algorithm 1

## C.1   Connection of the fixed-point method Newton's method

For a convex function $f : \mathbb{R}^p \to \mathbb{R}$, unconstrained Newton update on the parameter $\boldsymbol{\beta} \in \mathbb{R}^p$ reads

$$\boldsymbol{\beta}^{t+1} = \boldsymbol{\beta}^t - \left[\boldsymbol{\nabla}_{\boldsymbol{\beta}}^2 f(\boldsymbol{\beta}^t)\right]^{-1}\nabla_{\boldsymbol{\beta}}f(\boldsymbol{\beta}^t).$$

In our algorithm, we can easily see that

$$\boldsymbol{\nabla}_{\boldsymbol{\beta}}^2 f(\boldsymbol{\beta}^t) = \frac{1}{\mu}\sum_{i \in S^t}\mathbf{x}_i\mathbf{x}_i^T \quad\text{and}\quad \nabla_{\boldsymbol{\beta}}f(\boldsymbol{\beta}^t) = -\frac{1}{\mu}\sum_{i \in S^t}\mathbf{x}_i(y_i - \langle\mathbf{x}_i, \boldsymbol{\beta}^t\rangle) - \sum_{i \in [n]\backslash S^t}\mathbf{x}_i,$$

where $S^t = \{i \in [n] \; : \; y_i - \langle\boldsymbol{\beta}^t, \mathbf{x}_i\rangle \leq \mu\}$.

Replacing the Hessian with $X^TX = \sum_{i=1}^n\mathbf{x}_i\mathbf{x}_i^T$, we can write the update as

$$
\begin{aligned}
\boldsymbol{\beta}^{t+1} = & \boldsymbol{\beta}^t + \left[X^TX\right]^{-1}\left[\sum_{i \in S^t}\mathbf{x}_i(y_i - \langle\mathbf{x}_i, \boldsymbol{\beta}^t\rangle) + \mu\sum_{i \in [n]\backslash S^t}\mathbf{x}_i\right], \\
= & \boldsymbol{\beta}^t + \left[X^TX\right]^{-1}\left[\sum_{i \in S^t}\mathbf{x}_iy_i - \sum_{i \in S^t}\mathbf{x}_i\mathbf{x}_i^T\boldsymbol{\beta}^t + \mu\sum_{i \in [n]\backslash S^t}\mathbf{x}_i\right], \\
= & \boldsymbol{\beta}^t + \left[X^TX\right]^{-1}\left[\sum_{i \in S^t}\mathbf{x}_iy_i - X^TX\boldsymbol{\beta}^t + \sum_{i \in [n]\backslash S^t}\mathbf{x}_i\langle\mathbf{x}_i, \boldsymbol{\beta}^t\rangle + \mu\sum_{i \in [n]\backslash S^t}\mathbf{x}_i\right], \\
= & \left[X^TX\right]^{-1}\left[\sum_{i \in S^t}\mathbf{x}_iy_i + \sum_{i \in [n]\backslash S^t}\mathbf{x}_i\{\langle\mathbf{x}_i, \boldsymbol{\beta}^t\rangle + \mu\}\right],
\end{aligned}
$$

which reduces to our algorithm.

# D   Initialization routine for EXTRACT

We initialize EXTRACT also using robust estimation. First, we preprocess the movie matrix, and then we estimate cell sources one by one using Algorithm **??**. We list below the steps to our initialization routine in detail:

1. Spatially band-pass filter the movie, with the spatial kernel determined by an estimated average cell radius. Band-pass filtering is meant to smooth the activity enough to improve SNR, but also to get rid of any fluctuations in the low spatial frequency regime (such as large fluctuating background).

2. In a while loop:

   (a) Compute the maximum image of the movie in time dimension, and spot the maximum. Initialize a cell in the maximum location with a gaussian image kernel.

   (b) Use 1-component robust regression with the one-sided huber loss to estimate temporal and spatial bases in an alternating fashion. We solve each regression problem using the Newton's method since the hessian is a scalar and can be computed fast; furthermore, non-negativity constraint amounts to simple thresholding in this 1-component case. This step is typically very fast.

   (c) Add the found cell to the initialization set, and subtract the outer product of its image and trace from the pre-processed movie.

   (d) Continue until found maximums are lower than a set SNR, or number of found cells exceed a threshold.

We dispose of the pre-processed movie after intitialization in order not to introduce artifacts from spatial filtering, and stay consistent with the original calcium activity profile.

## E    Reproducing the Controlled Crosstalk Experiment with More Iterations

We carried out an experiment to reproduce Figure 3d when each algorithm is run for up to 6 iterations. Result is shown in Figure 1. The AUC gap between EXTRACT and CNMF is consistent with that seen in Figure 3d, and the gap looks even more pronounced for low fraction of initialized cells. This is an expected outcome and confirms the deterioration effect when using alternating estimation, which could be detrimental for non-robust estimators.

Figure 1: Reproducing the result in Figure 3d in the main text for 6 iterations.

## F    Reproducing Experimental Results under Different SNR Regimes

We re-ran the results in Figure 2d (see Figure 2) and Figure 3c (see Figure 3) from the main text. For the overlapping cells scenario, EXTRACT has increasingly better comparative performance agains CNMF with increasing SNR. For the experiment where we compare algorithms with and without neuropil, we observe somewhat intriguing outcomes. Particularly, CNMF deteriorates much faster than the other two algorithms with neuropil contamination when the SNR is decreased. EXTRACT keeps its relative performance with varying SNR, and all algorithms perform similar between neuropil and no neuropil cases when SNR is high.

## G    Runtime Comparison for EXTRACT, CNMF, and ICA

We compare the algorithms in the following setting: we fix the number of neurons in the FOV, and vary the FOV size. We do this by keeping the number of pixels equal to the number of time frames,

Figure 2: Reproducing the result in Figure 2d in the main text for different SNRs. EXTRACT outperforms CNMF in the AUC metric at various SNR levels.

i.e. maintaining a square movie matrix. For CNMF, we use the open-sourced MATLAB codebase maintained in the author's github page. For ICA, we use our internal version, built on the author's original open-source implementation. Our internal version optimizes matrix operations to speed up the computation of the ICA step. Consequently, the ICA mathod spends majority of its computation time at the PCA step, particularly for computing the eigenvalues of the covariance matrix of the movie using eigs() command. For EXTRACT, we use our MATLAB implementation which computes robust estimation on the GPU. We use NVIDIA GTX 1080 for the experiment. We perform 10 alternating estimation iterations after initialization. We show in Figure 4 the results of running two sets of experiments for 30 cells and 70 cells. We report the mean runtime over 10 experiments. EXTRACT performs very similar to ICA, and the two algorithms are notably faster than CNMF. As the size of the dataset increases, EXTRACT becomes faster than ICA.

Figure 3: Reproducing the result in Figure 3c in the main text for different SNRs.

Figure 4: Runtime comparison among the three algorithms. Error bars are 1 s.e.m.