[Reviews · NeurIPS 2017]

Reviewer 1



This paper proposes a robust statistical estimator for analyzing calcium imaging data to extract timecourses and spatial filters of cells. This is an important application for neuroscience and improvements that increase the ability of algorithms to deal with cross-talk would be useful for the field. The framework appears to be closely related to the CNMF method, but with a robust loss function. I'm not confident in assessing the novelty or validity of the statistical framework, but I will comment on the practical application. My main reservation is that in Fig. 2d it appears that the estimation accuracy declines with each iteration. I guess this is because the authors initialized the algorithm using the ground truth cell locations. This seems to be an unrealistic scenario, and is not justified in the paper. Given this, it's hard to evaluate the claim that the EXTRACT algorithm improves performance relative to CNMF (or ICA). Additional points: - Line 47: The reason that signals in calcium imaging data are non-negative is not because they come from photons. Rather, it is because calcium is highly regulated in neurons, with near zero baseline concentration of free calcium and several orders of magnitude increase in calcium following opening of voltage gated ion channels. - Why not simulate the data using a Poisson distribution to model photon shot noise? This is the most realistic noise model for fluorescence imaging data.

Reviewer 2



The authors contribute a matrix factorization method based on a one-sided Huber loss within the framework of robust estimation. Asymptotic proofs of optimality are provided. The method is applied to the matrix factorization problem of extracting cells and time courses from optogenetic neural measurements and improvements over the matrix factorization method CNMF, as well as ICA. As I am not familiar enough with robust regression I cannot comment much on the proofs. I believe the authors that this contribution is novel and that the proofs are correct. In this setting, the contribution is clear a step ahead in that it provides the possibility to perform the matrix factorization needed for the calcium imaging application. It is by construction more accommodating of otherwise disruptive types of noise and it is conceivable that this gives the method an edge over existing varieties. In the evaluation on synthetic data, possibly some more systematic variations of overlap/temporal similarity could have been explored and the outcome compared against the other methods in order to understand breaking points of the proposed method as well as the comparisons more systematically.

Reviewer 3



This is an important and timely paper in the automated signal detection of calcium imaging of neurons. The authors have developed a new methodology based on careful noise model and robust statistics. The prevalence of calcium imaging experimental studies has increased interest in powerful analysis methods for understanding correlation of neuronal firing patterns, and this paper represents a strong advance in this direction. The use of robust location estimator appears be a good approach, particularly given the magnitude of signal and noise variability and strong presence of ouliers in neuropil. It is somewhat surprising that previous methods have not considered this. The author's noise model as a superposition of positive sources and lower amplitude normal component is interesting and gives more statistical power in the analysis. In equations (2) these components should be more explicitly labelled for readability. The authors present a rigorously argument for the algorithms convergence rate which is fast. The authors present a nice comparison of their EXTRACT algorithm with two other approaches in actual manually labelled microendoscopic single-photon imaging data and show superior performance. This section of the paper could be developed a little more carefully with more explicitly stated performance statistics, although Figure 4 is well done. In summary this is a very strong paper on a timely and important topic. The availability of a code and how it might be deployed would be helpful.